# Myocardial Function after Coronary Artery Bypass Grafting in Patients with Preoperative Preserved Left Ventricular Ejection Fraction—The Role of the Left Ventricular Longitudinal Strain

**DOI:** 10.3390/medicina59050932

**Published:** 2023-05-12

**Authors:** Vasil Papestiev, Sasko Jovev, Petar Risteski, Aron Frederik Popov, Marjan Sokarovski, Valentina Andova, Ljubica Georgievska-Ismail

**Affiliations:** 1University Clinic for Cardiac Surgery, Faculty of Medicine, Ss. Cyril and Methodius University of Skopje, 1000 Skopje, North Macedonia; 2Department of Cardiac Surgery, University Hospital Zürich, 8091 Zürich, Switzerland; 3Department of Cardiothoracic Surgery, Helios Klinikum Siegburg, 53721 Siegburg, Germany; 4University Clinic of Cardiology, Faculty of Medicine, Ss. Cyril and Methodius University of Skopje, 1000 Skopje, North Macedonia

**Keywords:** coronary artery bypass grafting, coronary artery disease, echocardiography, speckle-tracking imaging, global longitudinal strain

## Abstract

*Background and Objectives*: The role of coronary artery bypass grafting (CABG) on postoperative left ventricular (LV) function in patients with preoperatively preserved left ventricular ejection fraction (LVEF) is still being discussed and only a few studies address this question. This study aimed to assess LV function after CABG in patients with preoperatively preserved LVEF using left ventricular longitudinal strain assessed by 2D speckle tracking imaging (STI). *Materials and Methods*: Fifty-nine consecutive adult patients with coronary artery disease (CAD) referred for a first-time elective CABG surgery were enrolled in the final analysis of this prospective single-center clinical study. Transthoracic echocardiography (TTE), with conventional measures and STI measures, was performed within 1 week before CABG as well as 4 months after surgery. Patients were divided into groups based on their preoperative global longitudinal strain (GLS) value. Differences in systolic and diastolic parameters between groups were analyzed. *Results*: Preoperative GLS was reduced (GLS < −17%) in 39% of the patients. Parameters of systolic LV function were significantly reduced in this group of patients compared to the patient group with GLS% ≥ −17%. In both groups, 4 months after CABG there was a decline in LVEF but statistically significant only in the group with GLS% ≥ −17% (*p* = 0.035). In patients with reduced GLS, there was a statistically significant postoperative improvement (*p* = 0.004). In patients with preoperative normal GLS, there was not a significant change in any strain parameters after CABG. There was an improvement in diastolic function parameters measured by Tissue Doppler Imaging (TDI) in both groups. *Conclusions*: There is improvement in LV systolic and diastolic function after CABG in patients with preserved preoperative LVEF measured by STI and TDI. GLS might be more sensitive and effective than LVEF for monitoring improvements in myocardial function after CABG surgery in patients with preserved LVEF.

## 1. Introduction

Coronary artery disease (CAD) is one of the leading causes of death worldwide and its incidence will only grow by the end of this decade [1]. Coronary artery bypass grafting (CABG) is a proven method for the treatment of CAD. It has been shown to improve symptoms, prevents sudden cardiac death, and improves prognosis in patients with left main and three vessel disease [2]. The role of CABG on postoperative left ventricular (LV) performance is still under debate. Many studies showed the beneficial effects of CABG on postoperative LV performance in patients with impaired preoperative LV function and ejection fraction (EF) below 50% [3,4]. On the other hand, only a few authors studied changes in LV function after CABG in patients with normal baseline LV function and EF > 50% [5,6]. In their studies, they showed a small but statistically significant reduction in LVEF after successful CABG. Several mechanisms have been hypothesized for this outcome and the question arises whether this method is sensitive enough to assess myocardial performance before and after CABG in patients with normal baseline LVEF. Although LVEF is the most common echocardiographic measure of LV contractility, it is a robust measure and some patients with hidden dysfunction may have a normal LVEF measured by two-dimensional (2D) echocardiography [7]. Speckle-tracking imaging (STI) is a novel sonographic technique designed to track the motion of myocardial tissues frame by frame in the form of 2D speckles in the region of interest [8]. Speckle-tracking echocardiography and its most important measure, global longitudinal strain (GLS), can distinguish between normal and abnormal myocardial deformation and detect subclinical LV dysfunction [9]. Therefore, it can be valuable in detecting preoperative contractile dysfunction in patients with good baseline LVEF referred for CABG. Also, this method can be shown to be valuable in the postoperative monitoring of myocardial function. Pre- and postoperative myocardial performance are two the most important harbingers for mid- and long-term major adverse cardiac and cerebrovascular events (MACCE) and long-term survival [10]. Monitoring of changes in postoperative myocardial function is important for prognosis and for further diagnostic and therapeutic procedures. That is why clinicians need a method that will be more subtle, reproducible, and less operator dependent than conventional echocardiography. On the other hand, LV diastolic function is still an underrated method for following LV performance after revascularization, although it is one of the first parameters for subclinical ischemia [11]. The role of pre- and postoperative diastolic function in CABG surgery and its significance have yet to be determined. The objective of this study was to assess LV function after CABG in patients with preoperatively preserved LVEF using conventional echocardiographic and 2D speckle tracking imaging (STI) methods.

## 2. Materials and Methods

### 2.1. Study Patients

This was a prospective single-center clinical study approved by the Medical Ethics Committee of Medical School, University Ss. “Cyril and Methodius”, Skopje (ethical approval code No 03-6116/2, approved on 12 November 2018). During the period of November 2018 to October 2019, after providing informed consent, we enrolled 65 consecutive adult patients. All patients were adults with stabile CAD and referred for a first-time elective CABG surgery. Exclusion criteria were urgent, emergent or salvage surgery; pre- or postoperative need for mechanical circulatory support (intra-aortic balloon pump or extra corporeal life support); any form of heart valve disease, regardless of severity; no associated surgical procedures, such as valve replacement or surgery of the ascending aorta; redo CABG; Patients with severe chronic kidney disease and patients on dialysis; patients with a severe form of chronic pulmonary disease; technically inadequate echocardiographic window; failure to appear for control examination; and lethal outcome during study period.

Acetylsalicylic acid was not discontinued before revascularization and it was continued in the first 6 h after the procedure. The second antiplatelet agent (clopidogrel) was administered between the 3rd and 6th postoperative day (except in three patients, where it was contraindicated). All patients received guideline directed medical therapy (statin, beta-blockers (BB), and angiotensin-converting enzyme inhibitors or angiotensin receptor blockers) at discharge. Our standard protocol was on-pump surgery with cardiopulmonary bypass (CPB) with the use of cold-blood cardioplegia. In every case, except one, the left internal thoracic artery (ITA) was used to bypass the left anterior descending artery (LAD). A saphenous vein graft was used for the revascularization of other diseased vessels. None of the patients received contralateral ITA or other arterial grafts as a second or third graft. A four-month follow-up was performed in 59 patients. One patient died in the early postoperative period and five patients did not appear during the control visit.

### 2.2. 2-D Echocardiography Parameters

Transthoracic echocardiography (TTE) was performed within 1 week before CABG as well as 4 months after surgery. We chose a four-month follow-up period because of data provided by Haas et al. [12]. In their study, no further LV recovery measured by TTE was seen 3 mounts after CABG. All measurements were made by a single experienced investigator on commercially available equipment (Vivid 7; General Electric, Milwaukee, WI, USA) according to the professional association recommendations [13]. LV dimensions in systole and diastole as well as septal and posterior wall thickness were measured from parasternal M-mode recordings. LV volumes normalized to the body surface area (BSA) and LV ejection fraction (LVEF %) were calculated using biplane method of discs (modified Simpson’s rule). LV mass was calculated using the linear method and was normalized to the BSA. For LV wall motion assessment, the 17-segment model was used and the score index was calculated (WMSI) in accordance to standard formula. Mitral annular plane systolic (MAPSE) excursion was evaluated using M-mode TTE by measuring the displacement of the mitral annulus in relation to the LV apex at all four walls and its average value was calculated. Left atria volume normalized to the BSA (LAVI) was measured using area-length method in apical four-chamber and apical two-chamber views at end-systole. Mitral flow parameters using PW-Doppler were measured (early and late diastolic flow and its ratio). Pulse wave-tissue Doppler imaging (PW-TDI) was performed in the apical four-chamber view to assess annular systolic (s’TDI) and early diastolic (e’) velocities at the septal and lateral wall. The ratio of mitral flow early diastolic wave (E) to e’ for each of these annular velocities (septal/lateral) and its average value was calculated.

### 2.3. 2D Speckle Tracking Echocardiography

The LV apical long, four- and two-chamber images at frame rates between 55 and 80 frames/s were used for assessing 2D speckle tracking LV longitudinal strain. Commonly three consecutive heart cycles were recorded. Global peak systolic longitudinal strain (GLS) was derived from the average value of 17 segments and each segment was analyzed individually. After the segmental tracking quality was assessed and eventually manually adjusted again, recordings were processed using acoustic-tracking software (Echo Pac, General Electric, San Ramon, CA, USA), allowing offline semi-automated analysis of speckle-based strain. Only myocardial segments considered to be of adequate quality by both the automatic system and the operator were included in the analysis. In patients in whom some segments were excluded because of the impossibility of achieving adequate tracking, calculations were done by averaging values measured in the remaining segments. A GLS of −17% was taken as the lower limit of normal according to the professional association recommendations [14] as well as specific vendor (GE) and software that were used.

### 2.4. Statistical Analysis

Categorical parameters were summarized as percentages and continuous parameters as mean ± SD. Comparisons of preoperative vs. postoperative data were performed using a Wilcoxon Signed Rank test for related samples. Continuous variables were compared using the nonparametric Mann–Whitney test for independent samples and categorical parameters were compared using Pearson’s chi-square test. Assessment of correlations was done using Pearson’s correlation analysis. All analyses was performed using SPSS version 25.0 (IBM SPSS, Inc., Chicago, IL, USA), and a *p*-value ≤ 0.05 was considered significant.

## 3. Results

### 3.1. Patient Characteristics

Fifty-nine patients with preoperative LVEF > 50% were included in the 4-month observational time and final statistical analysis. The patients were divided into two groups: group 1, those with abnormal preoperative GLS (GLS < −17%, *n* = 23), and group 2, those with normal preoperative GLS (GLS% ≥ −17%, *n* = 36). Baseline clinical and intraoperative characteristics are shown in Table 1 and were similar between groups. Seventy-two percent of the patients were male and there was no difference between groups. Patients with preoperative reduced GLS < −17% were more likely to have significant LM disease and a three-vessel disease but less likely to have chronic kidney disease. Most of the patients had hypertension and dyslipidemia, and nearly half were smokers. Two patients in group 1 and three patients in group 2 had atrial fibrillation (AF) preoperatively. At the four-month follow-up, there was only one patient with AF in each group. This distribution did not lead to a statistical difference between the groups. Cardiopulmonary bypass time and ischemic time did not differ between groups, and the majority of patients received three bypass grafts (median 3, range 2–5). In all patients, except one in group 2, a left ITA was used to graft the LAD artery (results not shown). 

### 3.2. Preoperative Echocardiographic Characteristics

Preoperative systolic and diastolic echocardiographic characteristics in patients and differences between groups divided according to the preoperative GLS are shown in Table 2. Except for parameters that show ventricular hypertrophy (PWd and IVSd), all other parameters of systolic LV function were significantly reduced in the patient group with GLS < −17% compared to the patient group with GLS% ≥ −17%. Regarding preoperative diastolic LV parameters, there was a statistically significant difference between the two groups only in the left atrial volume index (LAVI) and a borderline significance in the average diastolic mitral annulus velocity (e’ average).

### 3.3. LV Function after CABG in Patients Divided According to the Preoperative GLS%

Conventional echocardiographic systolic parameters and LV strain parameters in the study cohort as a whole and divided into groups of patients with normal and decreased preoperative GLS are shown in Table 3. When we analyzed the whole cohort, there was a statistically significant decline of LVEF by 3.3% and an improvement in GLS (*p* = 0.03). A statistically significant improvement in diastolic function parameters was also observed.

In both groups, there was a decline in LVEF after CABG, which was statistically significant only in group 2 (*p* = 0.035). As for the other conventional echocardiographic parameters, there was a significant improvement in LVIDs (*p* = 0.031), MAPSE (*p* = 0.031), and LVmass index (*p* = 0.024) in group 1, and a significant improvement in wall thickness parameters, IVSd (*p* = 0.007), and PWd (*p* = 0.043) and an improvement in MAPSE (0.035) in group 2.

In group 1 there was a statistically significant postoperative improvement in GLS (0.004) and a decrease in the number of segments with severely reduced LS < 13% by 2.4 (*p* = 0.005). In patients with preoperative normal GLS ≥ −17% (group 2) there was no significant change in any strain parameters after CABG.

When we looked at the diastolic function parameters in the whole cohort and in group 1, there was a statistically significant improvement in diastolic mitral annulus velocity (e’) measured by Tissue Doppler Imaging (TDI). In group 2 (GLS ≥ −17%) there were further improvement in E/e’ parameters.

Reproducibility: To assess the reproducibility as well as reliability of the LV strain measurements, we calculated the Intraclass Correlation Coefficient (ICC) by assessing 20 randomly selected images seen on two different occasions by the same investigator. The Intraclass Correlation Coefficient (ICC) for LV longitudinal strain measurements was 0.943 (95%CI 0.872–0.974).

## 4. Discussion

In our cohort, there was a statistically significant decrease in LVEF% (from 63.6 ± 8.5 to 60.3 ± 9.6, *p* = 0.018, an absolute decrease of 3.3%) after CABG. LVEF% remained in the reference range and so without clinical significance. Koene et al. [5] retrospectively analyzed the echocardiographic findings of 203 patients with preserved LVEF% before and after CABG. Although retrospective, this is the largest study to date analyzing pre- and postoperative echocardiograms in a population with preserved systolic function. Similar to our results, there was a statistically significant, but clinically insignificant, decrease in LVEF% (median decrease of 3%). On the other hand, Diller et al. [15] prospectively followed 32 patients for 18 months after CABG but did not find a significant reduction in LV systolic function after CABG. The same conclusion was reached by Yin et al. [6] in their series of 145 patients with preserved LVEF%. In their study, neither LVEF% nor the dimensions of LVDd and SVi changed after CABG. Flameng et al. [16] demonstrated a change in postoperative LVEF% in patients with preserved systolic function using PET scans. Patients with diffuse CAD in whom irreversible fibrotic changes of the myocardium have not occurred or in whom we do not have a large disparity between the demand and consumption of the myocardium, most often present with anginal complaints, without having an LV disorder. Especially in conditions of myocardial hypertrophy, sub-endocardial ischemia is the primary problem that is manifested by so-called “effort angina”. Such sub-endocardial ischemia, which affects the deep longitudinal muscle layer, in the long term leads to sub-endocardial fibrosis and gradual ischemic cardiomyopathy [17]. Sub-endocardial ischemia and subtle impairment of regional myocardial contractility are undetectable by conventional echocardiography, especially when performed at rest. Speckle tracking echocardiography, specifically global longitudinal deformation, should overcome this shortcoming of conventional echocardiography. It should provide an early indication of sub-endocardial ischemia and regional myocardial dysfunction [18,19]. In the entire cohort, there was a significant improvement in global LV strain after CABG (*p* = 0.03). This supports the findings of Yin et al. [6] and Durmaz et al. [20] who showed an improvement in the longitudinal deformation in patients with preserved LVEF after CABG. We believe that this finding is of exceptional clinical importance for two reasons: first, it confirms the success of myocardial revascularization and confirms the “awakening” of the subtly hibernating segments of the myocardium. Second, the method of echocardiographic assessment of strain, especially the measurement of GLS%, is shown to be superior to conventional echocardiographic measurements in the assessment of myocardial function after CABG in patients with preoperatively preserved LVEF%. However, GLS% must not be equated with LVEF% because the longitudinal myocardial deformation is mainly defined by the sub-endocardial layers. Impaired contractility in ischemic heart disease begins in the sub-endocardial layers. On the other hand, LVEF% gives us information about the middle myocardial layers (radial muscle fibers). All this explains the increasing value of GLS% versus LVEF% for detecting the earliest changes in myocardial function [9,21,22].

In our study, we analyzed patients divided according to GLS% because the longitudinal strain is taking an increasingly important role in clinical practice, especially in the evaluation of patients with ischemic heart disease and ischemic cardiomyopathy [22,23,24].

Myocardial dynamics are an extremely complex process and are not limited to systole and diastole. Longitudinal, circumferential, and radial movements, as well as torsional movements, define the complex revolution of the heart muscle. Disruption of one of these mechanisms will lead to dysfunction that is often undetectable by conventional measurements [25]. In our cohort, 23/39 of patients with a normal baseline LVEF had an impaired preoperative value of GLS%. The presence of hypertrophic myocardium [26] and dysfunction of the internal longitudinal muscle layer in conditions where the circumferential layer compensates [18,19] is part of the presumed mechanisms. Consequently, the question arises, which patients have “truly” preserved LV systolic function preoperatively and how should we recognize those patients? Such coexistence of reduced global myocardial deformation with normal LVEF% was described in the study by MacIver et al. [26]. This study explains the paradox of apparently preserved LVEF% and abnormal myocardial deformation in the presence of hypertrophic myocardium. However, in our study, due to the small number of patients in both groups, these two assumptions generate a hypothesis rather than a conclusion. Gozdzik et al. [27], in their study among 60 patients with preserved LVEF > 50%, 20% of patients had a preoperative impaired GLS, a finding that supports our findings. In these patients, LVEF% remained unchanged 6 months after CABG and, in our study, these patients had a decline in LVEF% post-CABG. When we analyzed these patients in our study, we obtained a postoperative improvement in LVDs and MAPSE but a decline in LVEF% of 2.5%, which was statistically insignificant. On the other hand, there was a statistically significant improvement in LV GLS% (*p* = 0.03) and a reduction in the number of LV segments with severely impaired longitudinal strain of 2.4 (*p* = 0.05), indicating improved regional contractility. These convergent results once again confirm the thesis that LVEF% and GLS% are not clinical synonyms but complementary parameters.

On the other hand, among all 36 patients with a normal pre-operative value of GLS ≥ −17%, all conventional parameters of LV systolic function were within reference values preoperatively. After successful revascularization of the myocardium, there was a reduction of the wall thickness (reduced stress conditions), statistically significant worsening of LVEF% (decrease of 3.6%, *p* = 0.042), and most importantly there was no change in GLS%. This is another indication that this group of patients had truly preserved myocardium preoperatively, which did not have the opportunity to improve postoperatively. In other words, normal systolic function after successful CABG remains normal. Finally, can we confirm that a normal GLS% value excludes any hidden myocardial dysfunction preoperatively? Diastolic mitral annulus velocity (e’) and E/e’ ratio, parameters measured by TDI, were suggested as the most important parameters of diastolic function by Swaminathan et al. [28]. These two parameters (e’ and E/e’) take an increasingly important place as precursors of LV dysfunction of ischemic origin, especially when the echocardiographic parameters of the systolic function are within the normal range [11,29]. In our study, patients in the group with preoperatively preserved GLS ≥ 17% had normal values of all conventional and strain echocardiographic parameters, but diastolic function parameters were below the reference values. Confirmation that diastolic dysfunction is a forerunner of systolic dysfunction in this group of subjects is the fact that these parameters improved statistically significantly after revascularization. Alternatively, if we define these findings differently, we can say that postoperative improvement of the diastolic function shown through the statistically significant improvement of TDI parameters and E/e’ ratio indicates successful revascularization.

In this study we aimed to understand the effect of surgical revascularization on basally preserved myocardium. Speckle Tracking Imaging and strain measurements should gain a role in the perioperative monitoring of cardiac function. That requires a larger number of studies with a larger number of patients, as well as the correlation of LV strain and parameters of diastolic function with postoperative angiography, positron emission tomography, and/or cardiac magnetic resonance, as well as with stress echocardiographic methods.

The major limitation in our study is that although all consecutive patients that met inclusion criteria were enrolled, we believe that the sample is too small and is thus hypothesis generating rather than definitive. Another disadvantage is that paired echocardiograms were done 4 months after CABG, a time that might be too short for complete myocardial recovery. Nevertheless, it revealed certain trends that might initiate corroboration in a multicenter larger population with a longer follow-up. We believe that single internal mammary artery revascularization is outdated and should be replaced with bilateral internal mammary artery or total arterial revascularization in most CAD patients. Although our study group was homogenous in terms of clinical and operative characteristics, we think that arterial revascularization might yield different and more conclusive results. 

## 5. Conclusions

Global longitudinal strain assessed by 2D speckle tracking imaging is a highly sensitive tool to obtain left ventricular systolic mechanics before and after CABG in patients with a preserved left ventricular systolic function. Postoperative improvement of GLS and parameters of diastolic function are findings that support the benefit of CABG on myocardial performance in patients with preoperatively preserved LVEF. Therefore, speckle tracking imaging might be more sensitive and effective than echocardiographic parameters, such as the LVEDd, LVEF, and stroke volume, for monitoring improvements in myocardial function after CABG surgery, especially in patients with preserved LVEF%. On the other hand, diastolic dysfunction might be one of the first signs of subtle myocardial ischemia, a fact that is often overlooked. Future studies correlating STI findings with postoperative angiography can establish this measurement as a prognostic parameter, directing further diagnostic and therapeutic methods accordingly.

## Figures and Tables

**Table 1 medicina-59-00932-t001:** Baseline clinical and intraoperative characteristics.

Parameter	All Patients *n* = 59	Group 1 GLS < −17% (*n*/%) = 23/39.0	Group 2 GLS% ≥ −17% (*n*/%) = 36/61.0	*p*
Age (years)	64.8 ± 7.5	65.6 ± 7.6	64.3 ± 7.4	0.750
Gender				
male/female (*n*/%)	43/16(72.9/27.1)	17/6 (73.9/26.1)	26/10(72.9/27.1)	0.154
BMI (kg/m^2^)	28.5 ± 4.5	28.9 ± 4.3	28.2 ± 4.7	0.613
Euro SCORE 2	1.6 ± 1.1	2.04 ± 1.40	1.17 ± 0.50	0.034
NYHA class	2.2 ± 0.5	2.3 ± 0.5	2.1 ± 0.5	0.314
Previous MI (*n*/%)	20/33.9	8/34.8	12/33.3	0.564
CKD (*n*/%)	9/15.2	2/8.6	7/19.4	0.017
AF (*n*/%)	5/8.5	2/8.6	3/8.3	0.553
Smoking (*n*/%)	27/45.8	9/39.1	18/50.0	0.292
Hypertension (*n*/%)	57/96.9	23/100	34/94.4	0.368
Dyslipidemia (*n*/%)	57/96.6	21/91.3	36/100	0.148
Diabetes mellitus (*n*/%)	24/40.7	11/47.8	13/36.1	0.423
SYNTAX score	30.9 ± 6.5	31.4 ± 7.8	30.7 ± 5.6	0.304
Left the main disease	21/35.5	12/52.2	9/25.0	0.039
Three vessel disease	48/81.3	21/91.3	27/75.0	0.081
Number of grafts	2.7 ± 0.8	2.7 ± 0.8	2.8 ± 0.7	0.538
CPB time (min)	105.9 ± 26.6	110.8 ± 26.8	102.8 ± 26.5	0.265
Ischemic time (min)	60.2 ± 16.3	61.5 ± 18.1	58.6 ± 15.5	0.363

AF = atrial fibrillation; BMI = body mass index; Euro SCORE 2 = European System for Cardiac Operative Risk Evaluation; MI = myocardial infarction; NYHA = New York Heart Association; CKD = chronic kidney disease; SYNTAX = SYNergy between percutaneous intervention with TAXus drug-eluting stents and cardiac surgery; CPB = Cardio Pulmonary Bypass.

**Table 2 medicina-59-00932-t002:** Echocardiographic parameters: comparison between groups.

Parameter	All Patients *n* = 59	Group 1 GLS < −17% *n* = 23	Group 2 GLS% ≥ −17% *n* = 36	*p*
LVIDd (mm)	49.0 ± 7.9	52.9 ± 7.8	46.5 ± 7.0	0.007
LVIDs (mm)	29.5 ± 8.4	33.9 ± 8.9	26.8 ± 6.9	0.003
IVSd (mm)	13.2 ± 2.2	13.1 ± 2.0	13.3 ± 2.0	0.609
PWd (mm)	11.6 ± 2.1	11.2 ± 2.5	11.8 ± 1.8	0.312
LVEDVI (mL/m^2^)	45.8 ± 15.9	55.1 ± 20.0	40.0 ± 8.7	0.002
LVESVI (mL/m^2^)	19.6 ± 16.1	22.9 ± 10.4	13.4 ± 4.8	0.001
LVEF (%)	63.6 ± 8.5	58.6 ± 7.0	66.8 ± 7.9	0.0001
MAPSEavarage (mm)	14.5 ± 2.3	13.5 ± 1.8	15.0 ± 2.4	0.001
s’TDI (cm/s)	6.4 ± 1.4	6.0 ± 1.5	6.7 ± 1.2	0.037
LVmass index	130.4 ± 38.2	141.2 ± 41.0	126.8 ± 37.4	0.019
WMSI	1.1 ± 0.2	1.29 ± 0.29	1.04 ± 0.07	0.0001
LV GLS (%)	−17.7 ± 3.9	−14.0 ± 2.6	−20.1 ± 2.6	0.0001
No segments with LV LS < 13%	3.8 ± 3.8	7.4 ± 3.6	1.5 ± 1.5	0.0001
LAVI (mL/m^2^)	33.4 ± 10.5	36.7 ± 10.1	31.3 ± 10.3	0.035
E/A	0.9 ± 0.5	0.8 ± 0.6	0.9 ± 0.5	0.231
e’ septal (cm/s)	5.3 ± 1.5	5.0 ± 1.2	5.6 ± 1.7	0.084
e’ lateral (cm/s)	7.5 ± 2.3	7.0 ± 2.5	7.8 ± 2.2	0.150
e’ average (cm/s)	6.5 ± 1.6	6.1 ± 1.7	6.8 ± 1.5	0.054
E/e’ average	11.2 ± 3.7	11.7 ± 4.6	10.9 ± 3.0	0.652

CABG = coronary artery bypass graft surgery; CI = cardiac index; e’ = diastolic mitral annulus velocity IVSd = septal wall thickness; LAVI = left atrial volume index; LVEDVI = left ventricular end-diastolic volume indexed to body surface area; LVEF = left ventricular ejection fraction; LVESVI = left ventricular end-systolic volume indexed to body surface area; LVIDd = left ventricular end-diastolic dimension; LVIDs = left ventricular end-systolic dimension; MAPSE = mitral annular plane systolic excursion; PAPs = pulmonary artery pressure systolic PW = posterior wall thickness; s’TDI = peak systolic mitral annular velocity by TDI; SVI = systolic volume indexed to body surface area; WMSI = wall motion score index.

**Table 3 medicina-59-00932-t003:** Echocardiographic parameters before and after CABG.

Parameter	All Patients *n* = 59			Group 1 GLS < −17% *n* = 23			Group 2 GLS ≥ −17% *n* = 36		
	Before CABG	After CABG	*p*	Before CABG	After CABG	*p*	Before CABG	After CABG	*p*
LVIDd (mm)	49.0 ± 7.9	48.5 ± 6.0	0.401	52.9 ± 7.8	50.7 ± 5.9	0.069	46.5 ± 7.0	47.2 ± 5.7	0.658
LVIDs (mm)	29.5 ± 8.4	28.3 ± 6.8	0.115	33.9 ± 8.9	31.0 ± 5.7	0.031	26.8 ± 6.9	26.5 ± 6.9	0.777
IVSd (mm)	13.2 ± 2.2	12.5 ± 2.4	0.026	13.1 ± 2.5	12.8 ± 3.1	0.951	13.3 ± 2.0	12.3 ± 2.0	0.007
PWd (mm)	11.6 ± 2.1	10.9 ± 1.9	0.039	11.2 ± 2.5	10.9 ± 2.0	0.531	11.8 ± 1.8	10.9 ± 1.9	0.043
LVEDVI (mL/m^2^)	45.8 ± 15.9	43.4 ± 14.4	0.167	55.1 ± 20.0	50.5 ± 16.1	0.191	40.0 ± 8.7	38.9 ± 11.2	0.545
LVESVI (mL/m^2^)	17.1 ± 8.7	19.6 ± 16.1	0.182	22.9 ± 10.4	22.3 ± 8.5	0.951	13.4 ± 4.8	17.9 ± 19.3	0.100
LVEF (%)	63.6 ± 8.5	60.3 ± 9.6	0.018	58.6 ± 7.0	56.1 ± 9.2	0.304	66.8 ± 7.9	63.1 ± 8.9	0.035
MAPSEav (mm)	14.5 ± 2.3	15.5 ± 2.1	0.002	13.5 ± 1.8	14.8 ± 1.8	0.031	15.0 ± 2.4	16.0 ± 2.0	0.035
s’TDI (cm/s)	6.4 ± 1.4	6.4 ± 1.2	0.918	6.0 ± 1.5	5.8 ± 1.3	0.697	6.7 ± 1.2	6.8 ± 0.9	0.577
LVmass index	130.4 ± 38.2	117.9 ± 34.7	0.020	146.1 ± 4.5	127.2 ± 41.6	0.024	120.5 ± 30.3	112.0 ± 28.6	0.272
WMSI	1.1 ± 0.2	1.1 ± 0.1	0.289	1.29 ± 0.29	1.18 ± 0.15	0.059	1.04 ± 0.07	1.05 ± 0.11	0.679
LV GLS (%)	−17.7 ± 3.9	−18.6 ± 4.3	0.030	−14.0 ± 2.6	−15.9 ± 3.6	0.004	−20.1 ± 2.6	−20.3 ± 3.9	0.652
No.seg.LVLS < 13%	3.8 ± 3.8	3.1 ± 3.3	0.073	7.4 ± 3.6	5.05 ± 3.9	0.005	1.5 ± 1.5	1.8 ± 2.0	0.565
LAVI (mL/m^2^)	33.4 ± 10.5	36.6 ± 8.6	0.023	36.7 ± 10.1	39.4 ± 7.5	0.181	31.3 ± 10.3	34.9 ± 8.8	0.065
E/A	0.9 ± 0.5	0.9 ± 0.4	0.360	0.89 ± 0.60	0.86 ± 0.31	0.927	0.90 ± 0.50	1.04 ± 0.56	0.010
e’ s (cm/s)	5.3 ± 1.5	6.2 ± 2.2	0.021	5.0 ± 1.2	5.1 ± 1.3	0.521	5.6 ± 6 ± 1.7	6.9 ± 2.5	0.020
e’ l (cm/s)	7.5 ± 2.3	10.5 ± 3.2	0.0001	7.0 ± 2.5	10.4 ± 3.3	0.001	7.8 ± 2.2	10.6 ± 3.1	0.0001
e’av. (cm/s)	6.5 ± 1.6	8.4 ± 2.2	0.0001	6.1 ± 1.7	7.7 ± 1.8	0.002	6.8 ± 1.5	8.8 ± 2.4	0.0001
E/e’ average	11.2 ± 3.7	10.0 ± 3.7	0.151	11.7 ± 4.6	10.7 ± 3.9	0.403	10.9 ± 3.0	9.6 ± 3.5	0.012

CABG = coronary artery bypass graft surgery; CI = cardiac index; e’ = diastolic mitral annulus velocity; e’av. = average; e’l = e’lateral; e’s = e’septal; IVSd = septal wall thickness; LAVI = left atrial volume index; LVEDVI = left ventricular end-diastolic volume indexed to body surface area; LVEF = left ventricular ejection fraction; LVESVI = left ventricular end-systolic volume indexed to body surface area; LVIDd = left ventricular end-diastolic dimension; LVIDs = left ventricular end-systolic dimension; MAPSE = mitral annular plane systolic excursion; PAPs = pulmonary artery pressure systolic.

## Data Availability

The data presented in this study are available on request from the corresponding author. The data are not publicly available due to privacy restrictions.

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
