# Peer review of "Myocardial Function after Coronary Artery Bypass Grafting in Patients with Preoperative Preserved Left Ventricular Ejection Fraction—The Role of the Left Ventricular Longitudinal Strain"

_medicina, 2023, doi:10.3390/medicina59050932_

Round 1

Reviewer 1 Report

Dear authors,

the review report is attached.

Author Response

Point 1: It is suggested to reformulate the title to: “Myocardial function after coronary artery bypass grafting in patients with preoperative preserved left ventricular ejection fraction – the role of the left ventricular longitudinal strain”.

Response 1: Dear Reviewer, we’ve reformulated the title in the original text according to your suggestion.

Point 2: There is no need for subheadings within the Introduction section.

Response 2: Dear Reviewer, the subheadings within the Introduction section have been removed.

Point 3: The overall English language of the manuscript should be improved. The grammar is overall fine, but the flow of multiple sentences should be improved. There is also use of wrong words (for example ‘mounts’ instead of ‘months’, etc.) Please read the manuscript thoroughly and revise where necessary.

Response 3: Dear Reviewer. We simplified the flow of some multiple sentences. We have changed the wrong words. Finally, the manuscript was sent to a professional proofreader for a final check

Point 4: The sample size of the study is limited given that 2 subgroups were compared.

Response 4: Dear Reviewer, we are aware that the number of patients enrolled in this study is too small and is thus hypothesis generating rather than conclusive. This is a point that we address in the limitation section. Off course more studies with more patients are needed, since Speckle Tracking Imaging (STI) is a relatively novel echocardiographic technic and its role in CABG surgery should be more extensively studied. We believe that this small group of patients gives us direction for future studies which will also correlate STI with early and long term outcomes in CABG patients.

Point 5: Full stop should be used instead of comma for decimal points in the results.

Response 5: Dear Reviewer, comma is replaced with full stop in the whole text

Point6: The results are unadjusted, and do not account for the potentially important differences between the groups, such as: completeness of revascularization, proportion of arterial revascularization, concomitant valvular disease, etc.

Response 6: Dear Reviewer, regarding clinical and intraoperative differences between groups:

  • Proportion of arterial revascularization: I agree that this is a very good point. During this study, the standard protocol for CABG patients at our clinic was LIMA to LAD and great saphenous vein to other diseased arteries. None of the patient in this study got contralateral internal mammary artery or radial artery as second or third graft. We will address your point in the “material and methods” section, but also in the “limitation” section.
  • Completeness of revascularization: Unfortunately we don’t have this data and it will be very difficult to extrapolate them now. In the results you can see that the Number of grafts (2.7 and 2.8) is same in both groups, and there is no difference in complexity of coronary artery disease between groups (Similar SINTAX score). That is why we don’t expect differences between groups regarding completeness of revascularization.
  • Concomitant valvular disease: none of the patients in this study had associated valvular disease. We added inclusion and exclusion criteria in the “material and methods” section to better clarify this isue
  • Preoperative atrial fibrillation (AF): Same percentage of preoperative AF in both groups. We added this data in the “results” section in order to show the homogeneity of the groups.

Taking into account all this preoperative demographic, clinical and intraoperative data we believe that this was a very homogenous cohort of patients without differences between groups

Point 7: The limitation section is inadequate.

Response 7: We change the text in this section and add more limitations that we believe are important to this study.

Additionally, we checked all references that were important to the article and made changes and described the research background and methods in more detail, according to your suggestion.

Reviewer 2 Report

The authors assessed LV function after CABG in patients with preoperatively preserved LVEF using left ventricular longitudinal strain (GLS) assessed by 2D speckle tracking imaging. They conclude that there is improvement in LV systolic and diastolic function after CABG in patients with preserved preoperative LVEF measured by STI and TDI.

The content is not special but valuable because this kind of small study contributes to later meta-analysis or reviews.

It is sometimes misunderstood that “GLS<-17” means the absolute value of GLS is larger than 17 (-18, -19…). The authors should state the definition of “GLS<-17” in this manuscript.

Reviewer 3 Report

This article evaluated how the echocardiographic parameters were changed after CABG in patients with preoperatively preserved LVEF. The global longitudinal strain was improved among the lower GLS group after a 4-month follow-up.

There are some points to be noted.

1 How did the authors evaluate medical therapies? The GDMT is vital to treat chronic coronary artery disease patients and may change the results of echocardiographic parameters due to its effectiveness.

2 Were patients with atrial fibrillation(AF) excluded from this analysis? AF could affect the echocardiographic parameters, and especially E/A could not be calculated in AF patients.

3 Though the authors mentioned that the e’ average was significantly different between the lower GLS group and the higher GLS group (line135), the p-value was 0.054, as shown in Table 2, which was not a significant difference.

4 Regarding the LAVI before CABG, why was LAVI larger in the GLS<-17% group than the GLS>-17% group? Did raising left ventricular end-diastolic pressure or atrial fibrillation affect it?

5 Comparing diastolic function parameters among all patients, whereas all kinds of e’ were improved, both E/A and E/e’ were not changed. What kind of mechanisms could be speculated about this result?

6 Though the authors concluded in group 2 that there was a statistically significant improvement in E/A, E/A > 1.0 is classified as a pseudo-normal pattern considering patients’ age. This means worsening left ventricular diastolic function.

7 What are the clinical implications of this analysis? What information was added to the current consensus?

Author Response

Point 1: How did the authors evaluate medical therapies? The GDMT is vital to treat chronic coronary artery disease patients and may change the results of echocardiographic parameters due to its effectiveness.

Response 1: Dear Reviewer. This point we will addressed in “material and methods” section. Dual antiplatelet therapy, high-intensity statin, beta-blockers and angiotensin-converting enzyme inhibitors or angiotensin receptor blockers is a standard post CABG therapy at our department.  

Point 2: Were patients with atrial fibrillation (AF) excluded from this analysis? AF could affect the echocardiographic parameters, and especially E/A could not be calculated in AF patients. 

Response 2: Dear Reviewer. Only 5 patients in total were in atrial fibrillation preoperatively (two in group 1 and 3 in group 2). Аt the four-month follow-up, there was only one patient with atrial fibrillation in each group. The analysis showed that this distribution did not lead to a statistical difference between the groups, so if there was an effect on the measurements it would be equally expressed in both groups and would not contribute to a significant difference in any measurements between them. According to your comment we have added this data to the “results” section

Point 3: Though the authors mentioned that the e’ average was significantly different between the lower GLS group and the higher GLS group (line135), the p-value was 0.054, as shown in Table 2, which was not a significant difference.

Response 3: Dear Reviewer. We’ve changed this in the “results” section

Point 4: Regarding the LAVI before CABG, why was LAVI larger in the GLS<-17% group than the GLS>-17% group? Did raising left ventricular end-diastolic pressure or atrial fibrillation affect it?

Response 4: Dear Reviewer. This is a good point. Since AF distribution between groups is non-significant, worse diastolic function with elevated end-diastolic pressure and worse myocardial contractility may explain this difference between groups

Point 5: Comparing diastolic function parameters among all patients, whereas all kinds of e’ were improved, both E/A and E/e’ were not changed. What kind of mechanisms could be speculated about this result?

Response 5: Dear Reviewer. The value of E/e' decreases (improvement of diastolic function) in all patients and separately in both groups, but statistically significant only in the group with GLS >17%. This may be due to lower myocardial stiffness in the group of patients with GLS >17%. For the E/A parameter we cannot give an explanation because, as an individual parameter it has no significance without also interpreting the Tissue Doppler parameters (e', E/e') and/or LAVI

Point 6: Though the authors concluded in group 2 that there was a statistically significant improvement in E/A, E/A > 1.0 is classified as a pseudo-normal pattern considering patients’ age. This means worsening left ventricular diastolic function.

Response 6: Dear Reviewer, as a single parameter for the interpretation of diastolic function, E/A is not significant. It is interpreted only together with the parameters we get from Tissue Doppler (e', E/e')... and LAVI

Point 7: What are the clinical implications of this analysis? What information was added to the current consensus?

Response 7: Dear Reviewer, we addressed this point in the “discussion” section

Round 2

Reviewer 1 Report

Thank you for the change. I have no more remarks.

Reviewer 3 Report

The authors revised this article precisely.